# Examining the Change of Human Mobility Adherent to Social Restriction Policies and Its Effect on COVID-19 Cases in Australia

**DOI:** 10.3390/ijerph17217930

**Published:** 2020-10-29

**Authors:** Siqin Wang, Yan Liu, Tao Hu

**Affiliations:** 1School of Earth and Environmental Sciences, The University of Queensland, St Lucia QLD 4067, Australia; 2Center for Geographic Analysis, Harvard University, Cambridge, MA 02138, USA; taohu@g.harvard.edu

**Keywords:** human mobility, COVID-19 spread, global pandemic, social restriction policy, Australia

## Abstract

The policy induced decline of human mobility has been recognised as effective in controlling the spread of COVID-19, especially in the initial stage of the outbreak, although the relationship among mobility, policy implementation, and virus spread remains contentious. Coupling the data of confirmed COVID-19 cases with the Google mobility data in Australia, we present a state-level empirical study to: (1) inspect the temporal variation of the COVID-19 spread and the change of human mobility adherent to social restriction policies; (2) examine the extent to which different types of mobility are associated with the COVID-19 spread in eight Australian states/territories; and (3) analyse the time lag effect of mobility restriction on the COVID-19 spread. We find that social restriction policies implemented in the early stage of the pandemic controlled the COVID-19 spread effectively; the restriction of human mobility has a time lag effect on the growth rates of COVID-19, and the strength of the mobility-spread correlation increases up to seven days after policy implementation but decreases afterwards. The association between human mobility and COVID-19 spread varies across space and time and is subject to the types of mobility. Thus, it is important for government to consider the degree to which lockdown conditions can be eased by accounting for this dynamic mobility-spread relationship.

## 1. Introduction

A novel coronavirus disease, called COVID-19 by the World Health Organisation (WHO), was first monitored and reported in December 2019 by the Chinese health authorities [1]. The outbreak of COVID-19 has spread across China and infected 72,436 Chinese people, of which 1868 died by 17 February [2]. Since 11 March 2020 when WHO announced the COVID-19 outbreak as a global pandemic, the confirmed COVID-19 cases have been spreading, with up to 118,000 cases in 114 countries, including Italy, Spain, Iran, the United States, Germany, France, and South Korea, which were the top eight infected nations in the early stages of the pandemic [1]. Australia, as an island country in the Global South, had its first confirmed COVID-19 case on 25 January 2020, a Victorian resident in their early 50s returning from Wuhan, China. COVID-19 spread rapidly in Australia to 298 confirmed cases by 15 March 2020 [3]. Since then, the Australian Government has implemented a series of “lockdown” policies to limit the transmission of COVID-19 infection by restricting human mobility, keeping social distances, shutting down local communities, and encouraging residents to stay at home with the exceptions of limited outdoor activities in local neighbourhoods [3]. As the growth rate of confirmed cases declined since early April and the growth curve further flattened in May, some national restrictions were lifted on 12 May. However, Australia experienced the second wave of the pandemic in mid-June due to international travellers coming back to Victoria. A series of local closure policies were re-introduced, mainly in Victoria and New South Wales, while other state’s borders remain closed.

With the two-wave pattern in Australia, a growing debate has emerged regarding the efficacy of the lockdowns, how much these policies affect human mobility, when and how the lockdowns should be eased, and whether it is possible to lift restrictions without unleashing additional waves of infection. An assessment of how the change of human mobility is adherent to social restriction policies and how mobility levels are associated with the growth rate of COVID-19 cases has been, and continues to be, urgently required and valuable in helping to navigate the policy dilemma and understanding the determinants of controlling the infection. In response to this pressing need, this study aims to achieve three objectives: (1) examining the temporal variation of COVID-19 spread and the change of mobility levels adherent to social restriction policies in eight Australian states/territories; (2) modelling and assessing the extent to which different types of mobility are associated with the COVID-19 spread; and (3) evaluating the time lag effect of mobility restriction on the COVID-19 spread.

To tackle these three objectives, we utilise data from the Google mobility report [4]. Google mobility data provides GPS-derived indices of how visits and length of stay at different places change compared to a pre-pandemic baseline, reflecting a behavioural switch to social restriction policies that are increasingly being implemented [4]. The advantage of Google mobility data is its large coverage of more than 130 countries at the national level and partially at the state or municipal level in some countries. Google mobility data has been used in some national-level studies e.g., [5,6,7,8], as well as studies at a smaller scale, revealing within-country variations over time e.g., [9,10]. In the context of Australia, there are limited empirical studies using survey data to investigate the relationship among transmission of COVID-19, social restriction policies, and human mobility [10,11]. In addition, Zachreson et al. used Facebook mobility data to produce spatial transmission risk assessment in local government areas in Melbourne [12]. However, the cross-state variation of the relationship between mobility change and virus spread is less explored, which is the focus of our study.

Methodologically, agent-based modelling has been employed in survey-based studies in the Australian context by using mobility and case confirmation data at the individual level [13,14]. For empirical studies at the aggregated population level, some common regression models that have been used in current scholarship include a negative binomial regression [15] with the dependent variable as categorized groups of the infected population due to the non-normal, highly-skewed, and non-continuous nature of COVID-19 data; a multiple linear regression model linking the number of daily confirmed cases to multi-dimensional variables [16]; and a panel regression model with consideration of a certain number of days after the case confirmation date [5]. The most relevant work to ours is the three-country comparison study by Tran et al., which focuses on the intensive lockdown period at the initial stage of the pandemic, with Australia as one of the case studies at the national level [7]. Employing a similar approach as used in their work, our study is central to the dynamics of the relationship between virus spread and human mobility at the state level across a longer timeline of the pandemic, emphasising the comparison of the first and second wave in Australia. Moreover, we also extend the examination of the time lag effect to include three timeframes (right after lockdown and 7 and 14 days after lockdown) to have a more holistic picture of the relationship between virus spread and mobility change.

Our analytical results show that the social restriction policies implemented in the early stage of the first wave controlled the COVID-19 spread effectively, and the restriction of human mobility has a time-lag effect on growth rates, which is more effective in controlling COVID-19 spread within seven days after the implementation of restrictions. In contrast, the types of mobility associated with the COVID-19 spread vary across the two waves and the Australian states/territories. We interpret these results with caution and link them to the findings from other countries for a discussion of policy implications.

## 2. Materials and Methods

We collected the daily confirmed COVID-19 cases from 15 February to 15 August 2020 from the Department of Health, Australian Government [3]. Mobility data were retrieved from Google COVID-19 Community Mobility Reports in the same period of time [4]. Google provides GPS-derived location information about the amount of time people spent in six types of locations, including workplaces, residential, parks, grocery and pharmacy, retail and recreation, and transit stations. Accordingly, mobility measures are divided into six categories with accessibility to these six types of locations, such as mobility to parks. Each type of data stream is encoded as a percentage change in the mobility metric, based on a baseline derived for the period of 3 January to 6 February 2020. These mobility data are regularly updated and released to the public for the express purpose of supporting public health bodies in their response to COVID-19. All of these datasets are fully anonymised and aggregated at the level of the Australian state/territory over each day.

Since 15 March 2020, the Australian Government has started to implement a series of travel restrictions, self-isolation, social distancing, and lockdown policies at the national level, with border closure policies implemented by state governments (Table 1). After the growth curve of COVID-19 flattened in May, some national restrictions were lifted on 12 May, but were reintroduced in mid-June in the state of Victoria due to the second wave of the pandemic. A series of local closure policies were implemented again, mainly in Victoria and New South Wales, while other states’ borders remain closed. Considering this unique two-wave pattern, our inspection of the temporal variation of COVID-19, mobility levels, and policy implementation at the first stage focuses on the full timeline from 15 February to 15 August, while the examination of the relationship between COVID-19 and mobility levels is divided into 2 periods, covering Australia as a whole and all states/territories in the 1st wave (15 March to 15 June) and focusing on Victoria as the main source of confirmed cases in the 2nd wave (15 June to 15 August).

Methodically, we first generated a combined mobility index (CMI) to represent the overall mobility change in a day compared to the pre-pandemic period, calculated as the mean of the mobility of each type i in a day t:(1)$$\mathrm{CMI}\left(t\right)=\frac{\sum_{i=1}^{6}{\mathrm{Mobility}}_{i}}{6}.$$

For COVID-19 cases, we examined its change over time alongside the implementation of the key policy interventions. Next, we calculated the growth rate and the doubling time of COVID-19 cases. Growth rate (percentage) at day t is calculated as [7]:(2)$$\mathrm{GR}\left(t\right)=\frac{C\left(t\right)-C\left(t-1\right)}{C\left(t-1\right)},$$
where C(t) is the cumulative number of confirmed cases at day t.

We also calculated the doubling time as another measure of virus spread. Compared to the growth rate as a percentage value, the doubling time as a typical epidemic measure indicates how long it will take for the infected population to double in size. It has also been commonly used in previous COVID-19 studies (e.g., [7]) to inform the impact of policy interventions on epidemic transmission, especially at the initial stage of the exponential growth. The doubling time (day) of confirmed cases at day t is calculated as:(3)$$\mathrm{DT}\left(t\right)=\frac{\mathrm{Ln}\left(2\right)}{\mathrm{Ln}\left(1+\mathrm{GR}\left(t\right)\right)}.$$

It has been acknowledged that the median incubation period was estimated to be 5.1 days, and 97.5% of COVID-19 patients develop symptoms within 11.5 days of infection [17]. Therefore, we selected three scenarios, accounting for the delays in reporting from the illness onset date, testing, and incubation over three time periods—right after the lockdown date, 7 days after the lockdown date, and 14 days after the lockdown date—to examine the relationship between mobility change and growth rate/doubling time of COVID-19 cases. We then conducted a correlation analysis between CMI and the growth rates/doubling time of COVID-19 cases and further investigated the association between each of the six types of mobility measures and growth rates/doubling time in Australia as a whole and in each state/territory via a series of ordinary least square regression models, accounting for the time lag effect:(4)$${\mathrm{COV}}_{s,t}=\alpha_{S}+\beta_{i}{\mathrm{Mob}}_{i,\;s,t-n}+\gamma_{t}+\varepsilon_{s,t}\;(n\;=\;0,1,\ldots ,t\;-\;1),$$
where COV denotes the growth rate/doubling time of COVID-19 cases in the state s on date t as described above;${\;\mathrm{Mob}}_{i}$ denotes each type i (i=1, 2…6) of mobility; and ${\mathrm{Mob}}_{i,\;s,t-n}$ is the mobility index in state s on the date (t − n). In this research, n equals 0, 7, and 14. $\beta_{i}$ is the standardised coefficient for each type of mobility; ε is the standardised error; $\alpha_{S}$ denotes the fixed place effect of state s and $\gamma_{t}$ denotes the fixed date effect for a transmission period after date t.

In our interpretation of the regression model, we emphasise the magnitude and significance of the coefficients, indicating the extent of the association between COVID-19 spread and the different types of mobility, rather than the fixed place effect and fixed date effect, denoting the variations of unobserved potential confounders underlying virus spread across space and time.

## 3. Results

### 3.1. The Change of Human Mobility and Policy Intervention

We first examine the two-wave pattern of the COVID-19 spread in Australia alongside the timeline of policy implementation (Figure 1). It is observed that COVID-19 cases increased exponentially from early March to 1 April 2020. Since 15 March, the Australian Government started to implement a series of travel restrictions, self-isolation, social distancing, and lockdown policies at the national level, including outdoor gatherings being limited to 500 people on 15 March; indoor gatherings being limited to 100 people on 18 March; further lockdown of restaurants, cafes, food courts, auction houses, and open house inspections on 26 March; and all gatherings of 2 persons only on 29 March. Simultaneously, Tasmania State Government closed its border on 19 March; New South Wales and Victoria shut down non-essential services on 22 March; Queensland, West Australia, South Australia, and Northern Territory State Government also implemented border closure policies on 24 March to prevent virus transmission across states. After almost two weeks since the first social restriction implemented on 15 March, daily confirmed cases reached a peak of 311 on 31 March as the turning point and started to decrease afterwards. With the prompt response of governments at the different levels and the control of human mobility, the daily confirmed cases have dropped from 611 on 23 March to 7 on 12 May. National restrictions were eased on 13 May and further eased on 1 June. However, daily confirmed cases started to increase again in mid-June, mainly caused by international travellers coming back to Victoria [18], leading to the second wave of the pandemic after three months of control. Victoria Government locked down 10 postcode areas on 30 June, applied Stage 3 Stay at Home restrictions on 8 July, and then upgraded to Stage 4 restriction on 2 August after it reached the peak of 717 daily confirmed cases on 30 July. Overall, the restriction policies implemented in the early stage of the 1st wave controlled the COVID-19 spread effectively. Although international travel led to the 2nd wave, domestic policies have been reintroduced to quickly intervene local transmission and reduce the daily confirmed cases to 293 on 15 August.

Following this timeline, we examine the average CMI within 3 days after the implementation of the key policies (Figure 2). The CMI in all states/territories have decreased consistently with the increasing restrictions of travel bans, social distancing, and self-isolation from 15 March to 2 April. After the turning point on 2 April, the CMI in all states/territories started to increase and the sharpest increase of mobility appeared in Northern Territory from 2 April to 12 May. After 1 June, the CMI of each state/territory varied over time. The human mobility in Tasmania remained in a relatively low level (e.g., around −20% on 30 June and afterwards) compared to the CMI in Northern Territory increasing from 0% on 15 May to 13.3% on 1 August. It may be possible that the weather in Northern Territory on winter days (May to August in Australia) is warmer and more inviting for outdoor activities compared to cold winter days in Tasmania, making people less active and mobile. It is noteworthy that the CMI in Victoria started to decrease after 21 June and so too for New South Wales and Australian Capital Territory after 8 July. This means that the re-introduction of restriction policies in Victoria responding to the second wave of the pandemic also affects the mobility in the adjacent state/territory.

We further examine each type of mobility in each state/territory (Figure 3). The pattern of mobility clearly presents a regular variation across weeks, evident as weekly cycles in most types of mobility with couples of “outliers”, indicating a substantial change of mobility on public holidays. Compared to the baseline (the period of 3 January to 6 February 2020 before the global COVID-19 pandemic), there are some common changes of mobility observed in all states/territories: the mobility to residence increased, indicating that more people stayed at home; the mobility to transit stations and retail/recreation decreased, reflecting that people used public transport and recreational facilities less; the mobility to workspace dropped substantially from 15 March to 15 April but started to increase from 15 April to 15 August, except in Victoria, indicating that people have been getting back to their work/business routines; the mobility to grocery/pharmacy substantially increased right after the implementation of the social restriction policy on 15 March, which coincided with the “panic-buying” where people stockpiled groceries and medicines to cope with the virus [19]. Finally, the mobility to parks varies across states/territories. The overall trend of mobility to parks decreased from 15 March to 15 April in all states but turned to increase slightly after 15 April. Compared to the time before the COVID-19 outbreak, most states have less mobility to parks from 15 March to 15 June, while the Northern Territory has more since May. Australian Capital Territory is observed to have a clear weekly circle with a substantial increase of mobility to parks over weekends. Some spikes are also observed in Victoria, New South Wales and South Australia on 14 June, Queen’s Birthday, reflecting the increase of park visiting on public holidays. It is noteworthy that the CMI in Victoria declined after 21 June as a result of the reintroduction of social restriction policies responding to the resurge of cases starting in early June. Then, the CMI in Victoria continued to decline but was accompanied by an increasing number of confirmed cases; such contrast trends reflect the complex relationship between virus spread and human mobility. It may be possible that the reintroduction of gathering limits in Victoria in early June was not restrictive enough to control the virus spread; Victoria did not completely lock down the hot spot, the inner city in Melbourne, until 2 August. In other words, the stringency and types of social restriction policy and the areas where policies were implemented collectively affect the virus spread. In sum, the implementation of social restriction policies at the early stage of the COVID-19 outbreak largely reduced human mobility to public facilities and spaces and such a reduction has been gradually eased with the lifting of restrictive policies, although the overall mobility during the COVID-19 outbreak remained lower than that in the period before the outbreak.

### 3.2. The Association between Human Mobility and COVID-19 Spread

Figure 4 shows the pattern of the CMI, growth rates, and doubling time of COVID-19 confirmed cases during the pandemic in Australia and in each state/territory. The growth rate of confirmed cases reached peaks in most states/territories (except Northern Territory and Australian Capital Territory) before 15 March when the social restriction policies started to be implemented. Two weeks of the increasing level of social restrictions from 15 March to 30 March (light and dark grey shadowed periods in Figure 4) largely reduced the growth rate and lengthened the doubling time of confirmed cases, accompanied by a substantial decrease of human mobility. The growth rate after 15 April remained at a low level in all states/territories except Victoria, where an increasing curve of growth rate appeared after 15 June but gradually flattened towards 15 August.

The correlation coefficients between CMI and growth rates in the first wave (15 March to 15 June) are significantly (*p* < 0.01) positive in most states except the Northern Territory (Figure 5A), indicating that a higher level of mobility is associated with a higher level of growth rates. The magnitude of correlation coefficients in most states increases from the period right after lockdown to the period seven days after lockdown, reflecting that the incubation period from 7 days to 14 days brings in a time-lag effect of human mobility on growth rates. However, the correlation between CMI and growth rates is insignificant in Victoria in the second wave (15 June to 15 August). Different from growth rates, the correlation between CMI and doubling time (Figure 5B) is insignificant in most states for both waves. The significant correlations only appear in three scenarios in Tasmania, while there is no substantial change of the magnitude of correlation over time. This indicates that the relationship between the doubling time and mobility varies across the timeline of the pandemic. This result is different from the findings by Tran et al. [7] where their report shows that the number of confirmed cases after the lockdown date was significantly associated with an increase in doubling time, particularly at the initial stage of the pandemic. This is possibly due to the fact that the number of confirmed cases at the initial stage tends to be in an exponential pattern but has more variations at the later stage of the pandemic, correspondingly being more difficult to explain by the change of mobility.

Figure 6 compares the regression coefficients of each type of mobility in each state/territory over three periods of time (right after, 7 days after the lockdown date, and 14 days after the lockdown date). In the first wave, the growth rate of confirmed cases has a negative association (significant at *p* < 0.01) with the mobility to retail/recreation only in the period right after lockdown (Figure 6A) and with the mobility to workspaces in most states only in the period in the period seven days and 14 days after lockdown (Figure 6B,C); while the mobility to transit stations is positively (significant at *p* < 0.01) associated with growth rates in all states across three periods of time and its magnitude increases over time (Figure 6A–C), indicating that a higher level of public transit usage is linked to a higher level of growth rates. Such a linkage becomes stronger after a seven day incubation period and slightly weaker after a 14 day incubation period, reflecting the time lag effect of mobility on COVID-19 spread and the delay of policy intervention.

Different to the first wave, the mobility to transit stations in Victoria in the second wave is significantly and negatively associated with growth rates across three periods of time (Figure 6A–C) and positively associated with doubling time (Figure 6D,E); while the mobility to workspaces is positively associated with growth rates only in the period right after lockdown (Figure 6A) but negatively associated with doubling time (Figure 6D). This indicates that growth rates are largely tied to the increasing level of people getting back to workspaces after three months of working at home in the second wave of the pandemic.

## 4. Discussion

Drawing on the COVID-19 data and Google mobility data, our study contributes an empirical study of the relationship among human mobility, social restriction policies, and COVID-19 spread in Australia. Due to the transmission dynamics and confounders underlying the epidemiological studies, we interpret our findings with caution and link them to the empirical experiences in other countries for a more holistic understanding of how human mobility intertwines with COVID-19 spread and for better policy implications.

First, a visual inspection of the COVID-19 cases and mobility level alongside the timeline of policy interventions in Australia suggests that social restriction policies controlled the COVID-19 spread effectively in the early stage of the first wave of the pandemic, during which the substantial decrease of human mobility as a consequence of the increasing level of social restriction was followed by a steep drop in growth rates and a sharp increase in doubling time of COVID-19 spread. This dramatic decline of growth rates could potentially reflect the fundamental association between the dynamics of the intense lockdown orders and virus transmission in the initial stage of the pandemic, also observed by [14]. The reduction of mobility has been gradually eased with the lifting of restriction policies in mid-May. However, the overall mobility still remained at a low level afterwards and has not been fully restored to the prepandemic level after the national restrictions were lifted. Moreover, there are also imperfect correspondences between social restriction and mobility levels to some degree, with mobility declining prior to formal restriction and in certain circumstances, increasing prior to formal restrictions easing and such observations have also been found in other countries including China, U.S., Sweden, and South Korea [7,20,21]. People may have intended to reduce access to public facilities and spaces with the precautions of virus spread before the implementation of social restrictions.

Second, the control of mobility has a time lag effect on COVID-19 spread as the span of the mobility-spread relation lasts from 7 to 14 days, which is possibly tied to the incubation period. We observe an increase in the strength of the mobility-spread correlation over the period from the time when restriction policies were implemented to 7 days after the policy implementation, but a decline in correlation from 7 days to 14 days after the policy implementation. There are more mixed patterns of mobility-spread correlation after the initial stage of intensive lockdowns. A possible explanation is that social restriction policies may influence virus spread not merely because of their direct effect on mobility levels, but also through their impact on other forms of individual behaviours, including individual social distancing, hygiene, and mask wearing [5]. The government level supervision and the efficacy of policy implementation, together with environmental conditions (such as changes in weather conditions) also affect growth rates in a manner that weakens the association between mobility and virus spread [14,22]. For example, the increase of mobility in Northern Territory after 1 April was followed by a well-controlled flat curve of growth rates compared to an obvious increase of growth rates in Tasmania in the same circumstance, possibly as the temperature on winter days in Northern Territory is much higher than that in Tasmania, which helps to control the virus spread.

Third, there exists a dynamic association between mobility in different types and COVID-19 spread, and the magnitude of such an association varies across space and time. In the first wave, growth rates were positively associated with the mobility to public transit and grocery/pharmacy in most states, but negatively associated with the mobility to retail/recreation and workspaces. With the growth of COVID-19 cases, people prefer to stay at home and avoid places of retail/recreation. In the meantime, they are required to work from home instead of workspaces. Such a finding has also been observed by Kissler et al. [23] in their study of New York City where the reduction in commuting movements is negatively correlated with COVID-19 prevalence. Furthermore, the mobility to public transit appears to be the only factor positively linked to the rise of growth rates over three periods of time in most states and such a linkage becomes stronger after the 7 day incubation period. However, different to the first wave, the mobility to public transit was negatively associated with the COVID-19 spread in the second wave of the pandemic in Victoria, where the rise in growth rates may be more subject to the increasing number of people moving back to workspaces after 3 months of working at home. This inconsistent relationship between mobility and COVID-19 spread reflects the fact that virus spread is not only relevant to variation in mobility levels, but is also subject to variation in other forms of preventative behaviours and perceptions, whether voluntary or government enforced [24]. Without consideration of the complexity of other potential confounders, which may have tangible and intangible impacts on COVID-19 spread, it would be arbitrary to conclude that any observed drop in growth rates is attributed to changes in mobility levels.

While the interpretation of our analytical results provides by no means definitive conclusions, it serves as an initial attempt that draws on publicly available measures of human mobility and COVID-19 data to study an epidemiological question with enormous social importance. There are certain limitations, imposing challenges in understanding the mobility-spread relationship. First, Google mobility data provides a relative measure of mobility change compared to the period from 3 January to 6 February 2010 as the baseline. The selection of the baseline may introduce some biases across different geographic contexts where human mobility may start to decline as an early reaction to COVID-19 and thus, it may not be representative of the prepandemic level. Second, the measure of CMI can be extended from the mean value as is used in this study to a weighted measure responding to the risk level of mobility by type. For example, based on the assumption that the mobility to the workspace is subject to a higher level of infection risk than the mobility to parks, we can assign higher weight to the former, which may result in a more realistic measure of CMI. Third, there are several types of delays that need to be considered in future analysis to reduce the data bias, including the delay between the date of the real-time mobility measure by Google data and the reported date of confirmed cases, the reporting delay from the illness onset date, and the delay further introduced by incubation and testing. Future work needs to examine over a longer period, ideally ranging from 1 day up to 21 days, with an incremental one day interval to provide a more detailed assessment of the time lag effect of mobility on virus spread. Moreover, the estimated growth rate from the incidence date should take into account the delay period to better quantify the impact of lockdown on the transmission dynamics. Fourth, further attention can be given to the exploration of the fixed place and date effect in the regression model to capture the temporal variation in potential confounders that may occur within geographic contexts over time. Since Google only provides state/regional level mobility reports, other data sources such as Geotagged Tweets can also help estimate human mobility changes in fine scale geographic areas [25,26,27]. Fifth, it is necessary to have a comparative study on the mobility-spread relation between Australia and those highly populated countries that have successfully contained the COVID-19 spread (e.g., South Korean, Japan and UK) for policy implications. For example, the Oxford COVID-19 Government Response Tracker (OxCGRT) systematically collects information on several different common policy responses that governments have taken to respond to the pandemic and proposes a stringency index for each country [28]. The quantitative policy index provides a new way to explore the associations among policies, human mobility, and the COVID-19 spread in future work.

Nevertheless, a great deal of caution must be exercised in understanding our findings, which may not be sufficient to indicate any causal direction between mobility control and virus spread. Although there is no straightforward way to infer policy prescriptions from our analytical results, we find suggestive evidence, which may help to mitigate virus spread, especially in the initial stage of the pandemic. First, the implementation of robust contact tracing systems and self-isolation within the 14 day incubation period would be crucial to attenuate the strength of the mobility-spread relation. Second, the dynamics between the initial lockdown and the later phase of the outbreak driven by individual behavioural changes reflect the importance of government level supervision and policy implementation should last for a longer period of time to maintain its efficacy. Third, as the span of the relationship between mobility and virus spread is suggested to be up to 14 days in our study, it is important for governments to consider the degree to which lockdown conditions can be eased by accounting for this window of time.

## 5. Conclusions

Drawing on data of confirmed COVID-19 cases and Google mobility data in Australia, we present a state level empirical study to examine how the change of human mobility is adherent to social restriction policies and how such changes affect COVID-19 spread. Our findings show that social restriction policies implemented in the early stage of the pandemic controlled the COVID-19 spread effectively, which largely reduced mobility levels. The overall mobility still remained at a low level afterwards and has not been fully restored to the prepandemic level after the national restrictions were lifted at a later stage. The restriction of human mobility has a time lag effect on growth rates in the initial stage as the strength of the mobility-spread correlation increases up to seven days after policy implementation but decreases afterwards. However, there are more mixed patterns of mobility-spread correlation after the initial stage of intensive lockdowns. The association between mobility and COVID-19 spread varies across space and time and is subject to the types of mobility. Thus, it is crucial for governments and policy makers to consider the degree to which lockdown conditions can be eased by accounting for this dynamic mobility-spread relationship.

## Figures and Tables

**Figure 1 ijerph-17-07930-f001:**
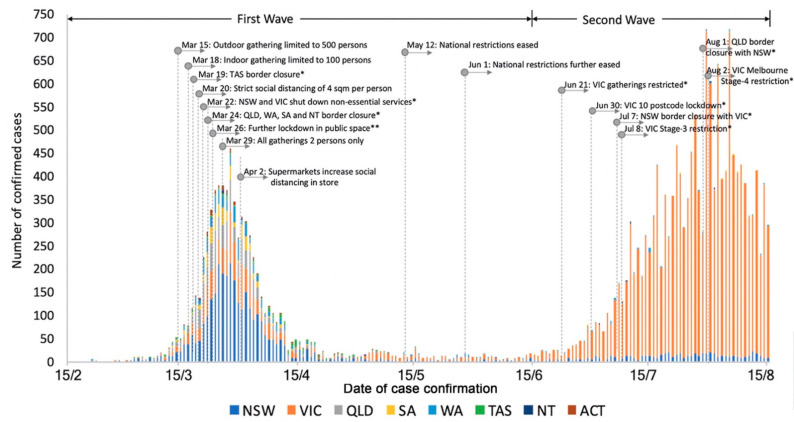
Confirmed COVID-19 cases in each state/territory and a timeline of policy intervention. Note: NSW: New South Wales; VIC: Victoria; QLD: Queensland; SA: South Australia; WA: West Australia; TAS: Tasmania; NT: North Territory; ACT: Australian Capital Territory; *: Policies at the state level; **: Further lockdown includes the closure of restaurants, cafes, food courts, auction houses, open house inspections; weddings restricted to 5 people; funerals to 10 people.

**Figure 2 ijerph-17-07930-f002:**
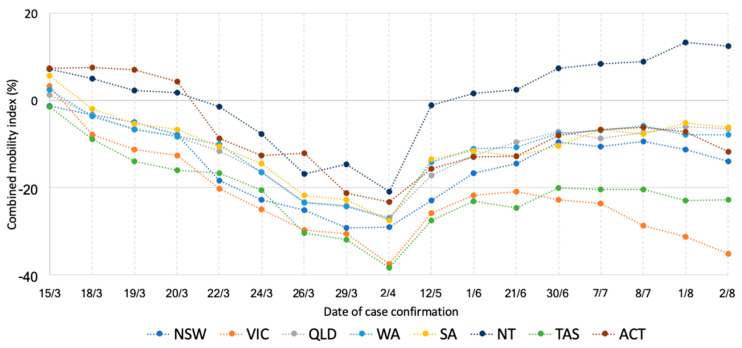
Combined mobility index (CMI) within 3 days after the implementation of each policy. Note: *italic* policies are at the state level. March-15: Outdoor gatherings limited to 500 persons; March-18: Indoor gatherings limited to 100 persons; March-19: *Tasmania border closure;* March-20: Strict social distancing of 4 sqm per person; March-22: *New South Wales and Victoria shut down non-essential services;* March-24: *Queensland, West Australia, South Australia, and Northern Territory borders closure;* March-26: Further lockdown of restaurants, cafes, food courts, auction houses, open house inspections; weddings restricted to 5 people; funerals to 10 people; March-29: All gatherings 2 persons only; April-2: Australian supermarkets increase in store social distancing measures; May-12: National restrictions eased; June-1: National restrictions further eased; June-21: *Victoria gatherings restricted;* June-30: *Victoria 10 postcode lockdown;* July-7: *New South Wales border closure with Victoria;* July-8; *Victoria Stage 3 restriction;* August-1: *Queensland border closure with New South Wales again;* August-2: *Victoria Melbourne Stage 4 restriction.*

**Figure 3 ijerph-17-07930-f003:**
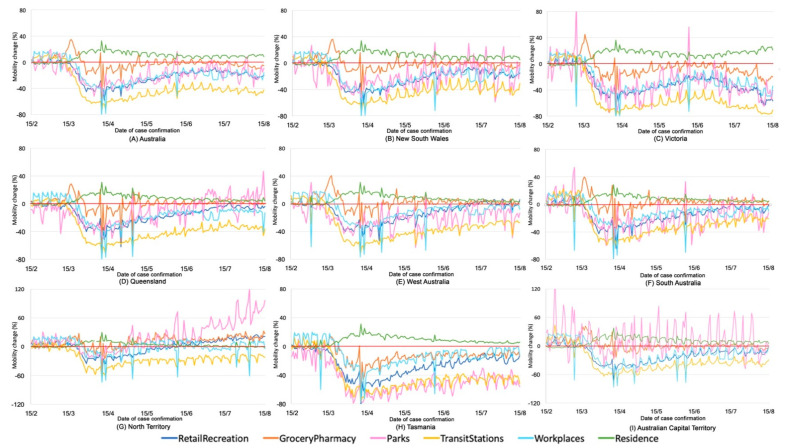
Six types of mobility in each Australian state/territory.

**Figure 4 ijerph-17-07930-f004:**
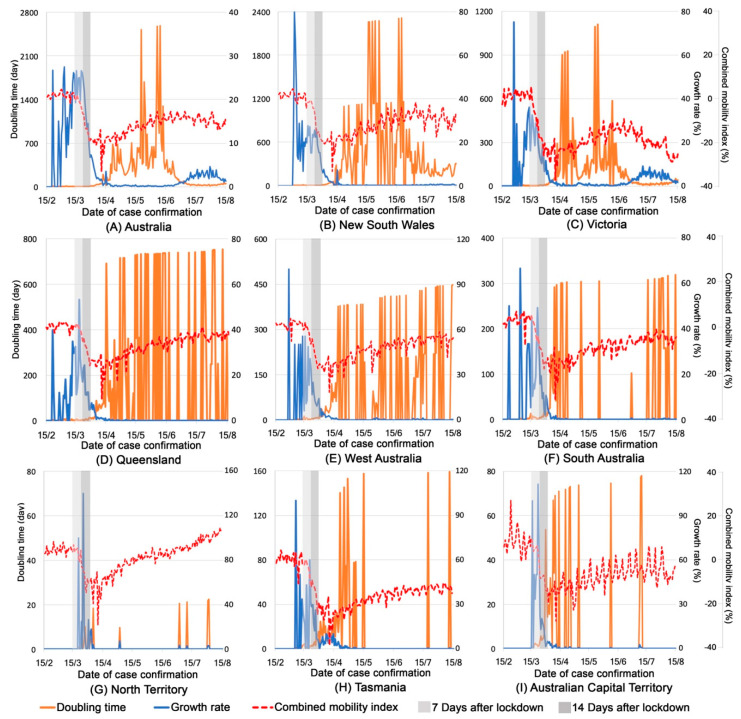
Combined mobility index, growth rates, and doubling time in each Australian state/territory. Note: In each graph, *Y*-axis on the left denotes doubling time (day); *Y*-axis on the right denotes growth rates (%); A combined mobility index is shown on the *Y*-axis on the right of each row.

**Figure 5 ijerph-17-07930-f005:**
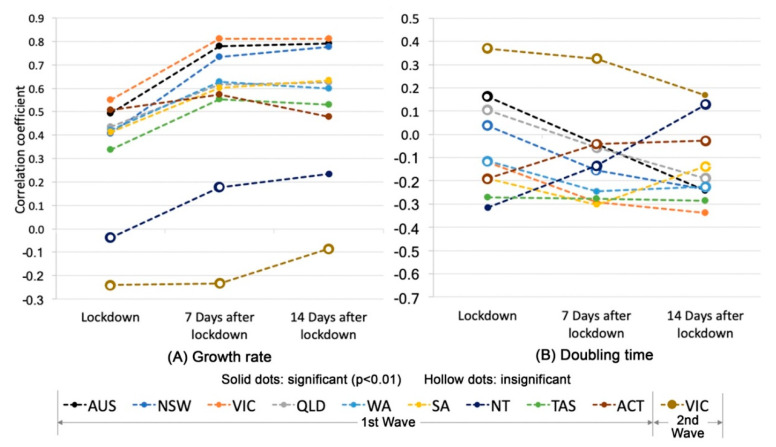
Correlation between CMI and growth rates (**A**), and between CMI and doubling time (**B**) in Australia and each state/territory over three periods of time. Note: the values of correlation coefficients are provided in Appendix A.

**Figure 6 ijerph-17-07930-f006:**
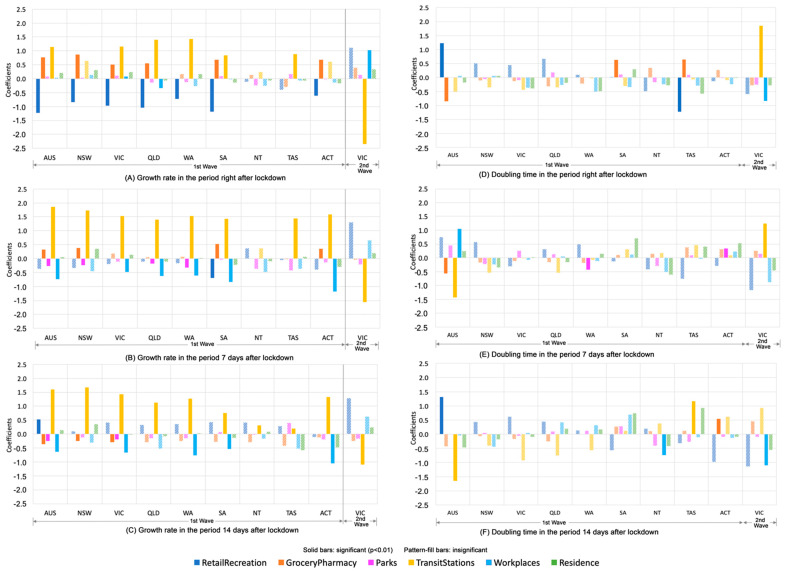
Regression coefficients of six types of mobility in Australia and each state/territory over three periods of time. Note: Regression coefficients are provided in Appendix A.

**Table 1 ijerph-17-07930-t001:** Social restriction policies implemented at the national and state level in Australia.

Date	Policy Restrictions
March-15:	Outdoor gatherings limited to 500 persons
March-18:	Indoor gatherings limited to 100 persons
March-19:	*Tasmania border closure*
March-20:	Strict social distancing of 4 sqm per person
March-22:	*New South Wales and Victoria shut down non-essential services*
March-24:	*Queensland, West Australia, South Australia, and Northern Territory border closure*
March-26:	Further lockdown of restaurants, cafes, food courts, auction houses, open house inspections; weddings restricted to 5 people; funerals to 10 people
March-29:	All gatherings 2 persons only
April-2:	Australian supermarkets increase in store social distancing measures
May-12:	National restrictions eased
June-1:	National restrictions further eased
June-21:	*Victoria gatherings restricted*
June-30:	*Victoria 10 postcode lockdown*
July-7:	*New South Wales border closure with Victoria*
July-8:	*Victoria Stage 3 restriction*
August-1:	*Queensland border closure with New South Wales again*
August-2:	*Victoria Melbourne Stage 4 restriction*

Note: *italic*: Policies are implemented at the state level; Source: [3].

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
