# Peer review of "Examining the Change of Human Mobility Adherent to Social Restriction Policies and Its Effect on COVID-19 Cases in Australia"

_ijerph, 2020, doi:10.3390/ijerph17217930_

Round 1
Reviewer 1 Report
The authors showed the impact of human mobility on the COVID-19 epidemiological situation in Australia. Overall, the article was written clearly. However, I have a major concern that would need to be addressed and it can be summarized as follow:
The human mobility data from Google is real-time, while the used epidemiological data is the number of confirmed COVID-19 cases by date of reporting, which might be partially biased by the reporting delay. Thus, the impact of changes of the mobility should be compared with the epidemiological curve by date of infection. Although the authors considered three different scenarios, I would be inclined to estimate the impact of mobility on adjusted the epidemiological curve by back-projection or simply moving back it with the delay (i.e., incubation period + reporting delay in Australia). Then, estimated values including the impact of mobility and time-lag may be changed. In addition, I would incline that authors should clarify clearly the x-axis of epidemiological curve (e.g., date of illness onset or date of confirmation).
Minor comment
1. Using the mean of each type of mobility in the calculation of CMI is totally logical. However, it would be interesting, if the authors can consider high- and low-risk mobility in terms of infection dynamics and give different weights on each type of mobility.
2. As shown in the Method section, the formula of doubling time already contains the growth rate. Thus, I would incline that authors can show only one indicator for representing the epidemiological situation.
Reviewer 2 Report
The methods, results and discussions are all clear and well-described. We can not deny this is a paper that deserves to be read. However, the authors should consider these minor and major remarks:
Minor revision
1. In the part of the paper directly before line 67, the authors did not firstly highlght more the advantage of the different uses of Google mobility data with respect to their subject, as we have some references like:
- Maloney, William, and Temel Taskin. "Determinants of social distancing and economic activity during COVID-19: A global view." (2020).
- Yilmazkuday, H. (2020). Stay-at-Home Works to Fight Against COVID-19: International Evidence from Google Mobility Data. SSRN. 3571708.
- Wang, H., & Yamamoto, N. (2020). Using a partial differential equation with Google Mobility data to predict COVID-19 in Arizona.
2. In the same part as in 1., the authors did not discuss the results of these following papers with respect to their regression models and they did not clarify how far the present method would be considered stronger.
- Oztig, L. I., & Askin, O. E. (2020). Human mobility and coronavirus disease 2019 (COVID-19): a negative binomial regression analysis. Public health, 185, 364-367.
- Cartenì, A., Di Francesco, L., & Martino, M. (2020). How mobility habits influenced the spread of the COVID-19 pandemic: Results from the Italian case study. Science of The Total Environment, 741, 140489.
Major revision
We also found this work:
Tran, T. H., Sasikumar, S., Hennessy, A., O’Loughlin, A., & Morgan, L. (2020). Interpreting the effect of social restrictions on cases of COVID-19 using mobility data. The Medical Journal of Australia, 1.
The authors should explain to us why their work should be considered different to that reference as there are enough similarities in the first part of methods and the country studied.
Reviewer 3 Report
This manuscript discusses the relationship between mobility and COVID-19 in different states of Australia. The authors explained the methods and the results scientifically, and they noted that although these results are not definite, they could be used for policymakers' initial decision. However, some issues should be addressed by the authors. Here are the problems:
- Line 62: "In within 7 days" is a typo? If it is not, please rewrite this sentence.
- Lines 70-71: different locations where people spent time are introduced, which later used as mobility type. It would be better if the authors mention that they are mobility types.
- Equation (4): C growth rate/doubling time while it is the cumulative number in Equation 3(which depends on t). It would be better if the authors choose another notation for growth rate/doubling time to avoid confusion.
- Lines 121-124: Why do the authors not interpret fixed place and time effect?
- Lines 137:141: There is a sharp increase in NSW in 15/4, which has not been discussed.
- Lines 142-143: The statement "Mainly caused by international travelers back to Victoria" needs proof. How the authors are sure, the second wave started by international travelers back to Victoria.
- Please make the manuscript body consistent with its Figure. If you use the abbreviations in the Figures, use them in the text as well; otherwise, it is better to apply the full names of states in Figures descriptions(Figure one and Two). I prefer the full name of states because many readers are not familiar with Australia states.
- In Figure 2 description, Please add "CMI" after the "Combined mobility index" at the description's start.
- Use the same colors for lines and bars in Figure 2, Figure 5, and Figure 1. I wanted to compare the two graphs, and it was confusing the colors of states were different in two figures. Generally, when the authors discuss states in plots, it would be better to use the same color for each state in all plots.
- In Figure 2, the CMI of Victoria State state is declined after June 21st, while the number of cases had an upward trend in the same state until early August. How do the authors explain these two different trends?
- Line 182: The baseline time interval is not presented in Figure 3. In figure 3, all plots are started on Feb 15th.
- The X-axis dates in Figure 3 are "day-month" while it is "Month-day." in Figure 2? Use consistent patterns in figures.
- Lines 190-191: "which was coincided with the reported ... with the virus" where is reported? Add reference or proof.
- Lines 231-248: Why doubling time plots has not been explained in Figure 6 for all the states? Only Victoria state in the second wave is discussed in Doubling time plots. Also, a table of regression results should be provided by the authors.
- Figure 6: Make the Y-axis range consistent in all plots. Use the same colors for each category in Figures 3 and 6.
- Why some of the bars are hatched in Figure 6?
- It would be better if the authors could specify those coefficients with p < 0.01 in Figure 6, as they specified them in the supplementary tables.
Overall, this paper presents an interesting idea, and the results validate the idea in most parts.
Round 2
Reviewer 1 Report
Overall, the manuscript is well revised. However, I have one more comment in the revised manuscript.
P11-12L268-372: Since the estimated growth rate rested on the incidence by reporting date, the time delay should include not only incubation period, but also reporting delay from the illness onset date to reporting (i.e., convolution of two delay distribution).
In addition, considering a longer time delay can be useful to overcome the uncertainty issue of time delay in this study. However, to quantify the impact of lock down on the transmission dynamics, the estimated growth rate from the incidence data accounted for the time delay should be explored. This part should be clearly addressed in the limitation section.
Author Response
Reviewer 1:
Overall, the manuscript is well revised. However, I have one more comment in the revised manuscript.
Response: Thanks again for the reviewer’s responsiveness and suggestions to improve the quality of our paper.
P11-12 L268-372: Since the estimated growth rate rested on the incidence by reporting date, the time delay should include not only incubation period, but also reporting delay from the illness onset date to reporting (i.e., convolution of two delay distribution).
Response: We have revised the method description to include ‘report delay from the illness onset date’ as the justification of setting up 3 timely scenarios (Line 136-137).
In addition, considering a longer time delay can be useful to overcome the uncertainty issue of time delay in this study. However, to quantify the impact of lock down on the transmission dynamics, the estimated growth rate from the incidence data accounted for the time delay should be explored. This part should be clearly addressed in the limitation section.
Response: We have revised the limitation to include this point suggested by the reviewer, and updated texts are read as (Line 369-377):
‘Third, there are several types of delays that need to be consider in the future analysis to reduce the data bias, including the delay between the date of real-time mobility measure by Google data and the reported date of confirmed cases, the reporting delay from the illness onset date, and the delay further introduced by incubation and testing. Future work need to examine over a longer period ideally ranging from 1 day up to 21 days, with an incremental one-day time lag to provide a more detailed assessment to the time-lag effect of mobility on virus spread. Moreover, the estimated growth rate from the incidence date should take into account the delay period to better quantify the impact of lock down on the transmission dynamics.’
Reviewer 2 Report
With these new discussions and references, the paper has become more interesting to read. The authors responded correctly to my suggestions, and the paper really deserves now to be published in IJERPH
Author Response
Thanks again for the reviewer's time and efforts to improve our paper.
Reviewer 3 Report
The authors addressed all of my concerns.
This manuscript presents an interesting idea that is supported by scientific evidence.
Author Response

(The authors gave the same response as above.)
